# Analysis of Common SNPs across Continents Reveals Major Genomic Differences between Human Populations

**DOI:** 10.3390/genes13081472

**Published:** 2022-08-18

**Authors:** Larisa Fedorova, Andrey Khrunin, Gennady Khvorykh, Jan Lim, Nicholas Thornton, Oleh A. Mulyar, Svetlana Limborska, Alexei Fedorov

**Affiliations:** 1CRI Genetics LLC, Santa Monica, CA 90404, USA; 2Institute of Molecular Genetics of National Research Centre, “Kurchatov Institute”, 123182 Moscow, Russia; 3Department of Medicine, University of Toledo, Toledo, OH 43606, USA

**Keywords:** computational biology, genomics, polymorphism, single nucleotide, genetic variation

## Abstract

Common alleles tend to be more ancient than rare alleles. These common SNPs appeared thousands of years ago and reflect intricate human evolution including various adaptations, admixtures, and migration events. Eighty-four thousand abundant region-specific alleles (ARSAs) that are common in one continent but absent in the rest of the world have been characterized by processing 3100 genomes from 230 populations. Also computed were 17,446 polymorphic sites with regional absence of common alleles (RACAs), which are widespread globally but absent in one region. A majority of these region-specific SNPs were found in Africa. America has the second greatest number of ARSAs (3348) and is even ahead of Europe (1911). Surprisingly, East Asia has the highest number of RACAs (10,524) and the lowest number of ARSAs (362). ARSAs and RACAs have distinct compositions of ancestral versus derived alleles in different geographical regions, reflecting their unique evolution. Genes associated with ARSA and RACA SNPs were identified and their functions were analyzed. The core 100 genes shared by multiple populations and associated with region-specific natural selection were examined. The largest part of them (42%) are related to the nervous system. ARSA and RACA SNPs are important for both association and human evolution studies.

## 1. Introduction

The 1000 Genomes Project revealed 81 million SNPs in humans [1]. Most of these SNPs present rare alleles with worldwide population frequencies less than 1% (68.4 million SNPs in 1000 genomes). Of the remaining SNPs, 2.7 million have alternative allele frequencies between 1 and 2%; 1.2 million between 2 and 3%; 0.76 million between 3 and 4%, and so on with fewer SNPs as allele frequencies increase. Due to the continuous sequencing of human genomes, the number of known SNPs constantly grows. In 2021, the NCBI public dataset of validated human SNPs was expanded to 1.053 billion [2], predominantly by inclusion of SNPs with very rare alleles. These rare SNPs are informative for inference of fine-scale population structure, investigations of admixture and migration events, and genome-wide association studies (GWAS) [3,4]. However, there are several limitations to using variants with rare alleles effectively. The first is sample size. To obtain appropriate statistical power, much larger sample sizes are required than in the case of common variants. The second limitation is the necessity of controlling for population ancestry, since populations differ in spectra of rare SNPs [5,6].

In contrast, common SNPs, with alleles that are more abundant (e.g., minor allele frequency (MAF) > 0.05), have been widely used in non-expensive and quick analyses in chip microarrays for different purposes. For example, ancestry explorations and finding close genetic relatives, and revealing the genetic bases of predispositions to common human diseases by GWAS. Alleles of common SNPs are often detected across all continents. However, their distributions have a large spectrum of patterns, varying from similar patterns through the geographic regions to pronounced local changes [7]. If population stratification of SNPs is not properly considered, incorrect conclusions may be made in GWAS. There are many publications on common SNP peculiarities in specific populations, including populations from different geographic regions and continents [8,9,10,11]. The most prolific among them utilize the genome-wide data of the International HapMap Project and the HGDP-CEPH Human Genome Diversity Cell Line Panel and whole-genome sequence data of the 1000 Genomes Project [12,13,14]. However, the population sets of the HapMap and 1000 Genomes projects suffer from two limitations: (1) samples from the Oceania region were not included and (2) American samples were admixed. For a long time, the HGDP-CEPH data included only genome-wide microarray genotypes and thus were not suitable for exploration of the full spectrum of SNPs. Recently the HGDP-CEPH populations have been sequenced; however, SNP distributions were not analyzed in detail [15]. Thus, the global picture of SNP distributions among continents and populations is not yet well described. The goal of this paper was to investigate extreme biases in alleles of common SNPs in different geographical regions and explore the biological reasons of these biases.

Common SNPs can be generally subdivided into “private” and “shared” on the basis of their occurrence in a single population or a range of populations. Occurrence of population-specific (population private) common SNPs has been previously investigated by Choudhury et al. (2014) [9] using the 1000 Genome dataset. These authors used a 5% cutoff for MAF to classify a variant as a common SNP. While Choudhury and co-authors state that the choice for this frequency cutoff for characterization of common SNPs is rather random and “does not have any special biological relevance,” it must be considered that common human SNPs with lower frequencies tend to be younger than SNPs with higher frequencies. Moreover, common human SNPs, characterized by much higher cutoff frequency (20–25% MAF), often form haplotypes with very specific and intriguing properties. These haplotypes are known as “yin yang” or mutually exclusive haplotypes [16,17]. These “yin yang” haplotype pairs become practically undetectable when the cutoff allele frequency drops down to 5%. Therefore, in this paper, we used a much higher cutoff frequency around 20%. Specifically, we chose 18% due to the small number of individuals from the examined geographical regions from the Simons Project of sequenced human genomes.

The results of Choudhury and co-authors are excellent and worthy of serious consideration. However, it is interesting to note that their data on population-specific common SNPs extensively differ from our region-specific common SNPs. This difference in results may be due to factors other than common SNP frequency cutoff. Additionally, in our study, neighboring mutually admixed populations were considered as belonging to the same geographical region. The idea of such an approach correlates with the results of Coop et al. (2009) [7], where the authors showed that even the SNPs with extreme *F_ST_*-values were distributed with high regularity across the geographic regions. This approach allows for the detection of common genetic characteristics that distinguish populations in one region from populations in other regions. The data on degree of population admixture were taken from a previous study [4].

In this paper, we characterized two types of region-specific common SNPs. The first type is abundant region-specific allele (ARSA) SNPs for which alleles are abundant (frequency > 18%) within a particular geographical region and almost absent in the rest of the world. The second type is regional absence of common allele (RACA) SNPs, which are widespread globally (allele frequency > 20%), but absent in one continent.

Thousands of region-specific common SNPs restricted to Africa, America, East Asia, Europe, and Oceania were revealed. We make them publicly available and encourage their use for science and industry.

## 2. Materials and Methods

### 2.1. Databases

The following whole genome sequence datasets were used for this paper: (1) the 1000 Genomes Project (phase III), including 2504 individuals from 26 populations [18]; (2) the Simons Genome Diversity Project, including 279 genomes from 142 populations [19]; (3) the Estonian Biocentre Human Genome Diversity Panel, including 402 new genomes from 125 populations [20]. The variant call format (VCF) file from each database was used to determine allele identification and count. Allele status (reference or alternative allele) was read from the genotype field (GT) of the VCF file. Initially, ARSA SNPs were identified from the Simons database and their alleles verified with the 1000 Genomes database and subsequently the Estonian HGDP database.

### 2.2. Computation of ARSA SNPs

A diagram for our approach in computing ARSA SNPs is shown in Figure 1. For computation of ARSA from the Simons database, we used the following thresholds for allele frequency: (1) African region: 17 or more counts of an Africa-specific allele out of 47 people from native African populations (Mozabite, Bantu, Biaka, Mbuti, Gambia, Luo, Masai, Luhya, Somali, Ju-hoan, Yoruba, Esan, Mandenka, Mende, Khomani-San, Dinka, Saharawi); 17 counts out of 47 people is equivalent to an 18.1% African ARSA frequency threshold. (2) American region: 3 or more counts of America-specific allele out of 8 people from Native American populations (Chane, Karitiana, Surui, Piapoco). In this case, the frequency cutoff threshold is 18.75%. (3) Oceanian region: 7 or more Oceania-specific allele counts out of 19 people from Oceanian populations (Australian, Bougainville, Papuan) with a threshold of 18.4%. (4) East Asian region: 8 or more East-Asia-specific allele counts out of 22 people from East Asian populations (Dai, Han, Japanese, Korean, Miao, Naxi, She, Tujia, Yi) with a threshold of 18.2%. (5) European region: 15 or more Europe-specific allele counts out of 41 people from 20 European populations (threshold = 18.3%). The small number of individuals in the Simons Database meant that the ARSA frequency cutoff threshold could only be set at discrete values. For example, the American ARSA threshold was 3 counts of American-specific alleles from 8 people (allele frequency cutoff threshold 18.75%). Increasing this threshold incrementally to 4 allele counts among 8 people would make this new threshold 25%. Therefore, the options for threshold count are naturally limited. The 18% threshold was optimal for all five regions and produced minimal deviation between regions (from 18.1% to 18.75% frequency cutoff threshold). During the second step, the identified ARSA SNPs were validated on the 1000 Genomes dataset. For the African region, we used six populations (YRI, LWK, GWD, MSL, ESN, and ACB) that are represented by 600 people. For the American region, we used four populations (CLM, MXL, PEL, and PUR) comprising 347 people. For the European region, we used 503 individuals from CEU, FIN, GBR, IBS, and TSI populations. For the East Asia region, we used 504 people from CDX, CHB, CHS, JPT, and KHV populations. Oceania does not have representatives in the 1000 Genomes dataset and was not tested for ARSA frequencies at this stage. For all regions, the ARSA frequency cutoff threshold was established exactly at 18.0%. Since many populations from 1000 Genomes are significantly admixed (especially American populations), the percentage of admixture was used to adjust cutoff frequency thresholds. For example, in a population with 40% admixture, the initial threshold of 18% was reduced by the amount of admixture, becoming 10.8% in this example. We used our previous publication to characterize admixture between populations from 1000 Genomes, the Simons, and the Estonian HGDP [4]. Data on percent admixture were obtained from Appendix A of the same publication. We also required that the frequency of ARSA in all other regions were at least 40 times less than in their region of specificity. The regions with significant admixture have been excluded from these calculations. For instance, in calculating the frequency of European-specific ARSA in the rest of the world, the South Asia (SAS) and American (AMR) populations were excluded. For Oceania-specific ARSA, we also used this control with more stringent conditions (less than 7 Oceania-specific allele counts were allowed among all individuals of 1000 Genomes dataset). Finally, in the third step, ARSA SNPs that passed the frequency test on 1000 Genomes were verified on the 402 genomes of the Estonian HGDP. The same criteria were applied: ARSA SNPs must be abundant in populations inside their specificity region and be practically absent in the rest of the world. After completing this step, the program generates comprehensive tables of ARSA occurrences in all tested populations from three genomic datasets. These five tables, namely, ARSAtableAFR, ARSAtableAMR, ARSAtableEAS, ARSAtableEUR, and ARSAtableOCE, are presented in the supplementary package and described in the ATLAS3protocols.docx file.

### 2.3. Computation of RACA SNPs

Calculations of RACA SNPs were performed with approaches very similar to ARSA computations but using only one step where only the 1000 Genome database was processed (see the Section 3).

### 2.4. Perl Programs and Computation Protocols

Computations of ARSA and RACA SNPs were performed by our pipeline of Perl programs specific to each continental region. All Perl programs are available on our website (http://bpg.utoledo.edu/~afedorov/lab/ATLAS3.html accessed on 20 May 2022) in a package that includes an Instruction Manual (ATLAS3instruction.docx) and Protocols (ATLAS3protocols.docx). In addition, this package of programs and protocols is available in Appendix A. Specifically, for obtaining African ARSA, we used the following programs: (1) AfricaSimonStep1.pl; (2) Africa1000gStep2_v3.pl; (3) AfricaSNPsESTONIA_v3.pl. For American ARSA—(1) AmericanSNPs2020.pl; (2) America1000gStep2_v3.pl; (3) AmericanSNPsESTONIA_v3.pl. For East Asia ARSA—(1) ChinaSimonStep1.pl; (2) China1000gStep2_v3.pl; (3) ChinaSNPsESTONIA_v3.pl. For European ARSA—(1) EuropeSimonStep1.pl; (2) Europe1000gStep2_v3.pl; (3) EuropeSNPsESTONIA_v3.pl. For obtaining Oceania ARSA, we used the following programs: (1) OceaniaSimonStep1.pl; (2) Oceania1000gStep2new_v3.pl; (3) OceaniaSNPsESTONIA_v3.pl. For obtaining African RACA SNPs, we used PopulationSpecific1000gAFRreverse.pl; for East Asia RACA SNPs—PopulationSpecific1000gCHIreverse.pl; for European RACA—PopulationSpecific1000gEURreverse.pl. The details of these computations are shown in ATLAS3_PROTOCOLS.docx Appendix A. In the output data files, we used the same identifiers for the individuals, populations, and geographic regions under analysis as in our previous publication [4].

Ancestral or derived status for the ARSA and RACA was calculated using VCF files of the 1000 Genomes Project. In column 8 of these VCF files, the ancestral allele is shown as “AA = n”, where n is the ancestral allele.

### 2.5. Characterization of Genes Associated with ARSA and RACA SNPs

The genomic context of ARSA and RACA SNPs was determined using the snpEff 5.1 program [21] with database GRCh37.87. Only canonical validated transcripts and variants intersecting them were considered. The intersections of gene lists were visualized with UpSet diagrams [22] that were built with the UpSetR package [23]. UpSet diagrams are equivalent to Venn diagrams but they are considered as being easier to read, specifically in the cases when intersections of many data sets (e.g., four and more ones) are analyzed.

## 3. Results

### 3.1. Characterization of Abundant Region-Specific Allele (ARSA) SNPs

We characterized abundant region-specific allele (ARSA) SNPs as those that alleles are abundant (frequency > 18%) within a particular geographical region and almost absent in the rest of the world. Some examples of ARSA SNPs are illustrated in Figure 2. It should be clarified that several regions/continents have strong genetic admixture with each other; therefore, for such regions, we introduced exceptions for ARSA SNP characterization as described in the Materials and Methods section. For example, in Figure 2A, African populations have noticeable admixture with Native Americans from the 1000 Genomes dataset (PEL, MXL, PUR, and CLM populations). Figure 2C reflects the highest known admixture between continents—Europeans vs. Americans and Europeans vs. Indo-European populations from South Asia (SAS group, from the 1000 Genomes Project). Therefore, our algorithms for identification of European ARSA require that their presence in Africa and East Asia should be 40 times less frequent than in Europe but allow noticeable presence of these alleles in America and South Asia.

ARSA SNPs were identified by consequentially processing three databases, starting with the Simons Human Genome Diversity Project. The initial ARSA SNP datasets obtained from the Simons database were then verified on the 1000 Genomes dataset, and finally on the Estonian Genome Diversity Project (EGDP) dataset. The advantage of the Simons project is that individuals with sequenced genomes represent pure populations with minimal admixture [4]. Its main disadvantage is the very small numbers of individuals representing each population. Counts of ARSA after Step-1 processing of the Simons database are shown in Table 1, column #2. During the second computational step, the frequency of ARSA in their specific regions (>18%) was confirmed on the largest dataset of 2504 genomes from 1000 Genomes Project. In addition, computations verified that ARSA frequencies are at least 40 times lower in the rest of the world. These data are displayed in Table 1, column 3. Note that Oceania populations are absent in 1000 Genomes; thus, the requirement for Oceania ARSA frequency > 18% is omitted at Step-2. However, the >18% frequency cutoff was enforced at the next Step-3 for the EGDP Database, which has 51 individuals from Oceania populations. Finally, at Step-3, we confirmed that (1) ARSA frequencies from Step-2 are also significant in their specific regions in EGPD database and (2) frequencies of ARSA in EGPD database are drastically reduced in other regions. The final numbers of verified ARSA are shown in Table 1, column 4, while the entire set of these SNPs is available in Appendix A. Many neighboring ARSA SNPs are in linkage disequilibrium with each other. Therefore, ARSAs from Step-3 were grouped into clusters where the distance between neighboring ARSA SNPs is less than 5 kb. The number of these clusters is shown in Table 1, column 5. The last column of Table 1 shows the proportion of ancestral versus derived alleles among ARSA. Interestingly, African ARSA have a significant proportion (22%) of ancestral alleles, much higher than the other four regions.

The highest number of ARSAs was detected in Africa, while the lowest number was observed in East Asia (Table 1). The second highest was in Native Americans, whose total number of ARSAs was nine times greater than the number of ARSAs from East Asia.

### 3.2. Characterization of Regional Absence of Common Allele (RACA) SNPs

We were also interested in the evaluation of SNPs with regional absence of common allele(s) (RACA). Examples of RACA distribution among populations are illustrated in Figure 3. In Figure 3A, RACA allele G is practically absent in the East Asia region, but very abundant in the rest of the world. In Figure 3C, RACA allele G is very rare in Europe but frequent in other continents. We allow for a very rare occurrence of RACA allele in their specific regions because of minor genetic admixture between continents. For example, South Europeans have higher genetic diversity, which has been associated with gene flow from Africa [24,25]. We computed RACA alleles only from the 1000 Genomes dataset because absence of an allele in a region can only be proven using larger populations of hundreds of individuals. In other words, it is not reasonable to assume population-wide absence of an allele in small population datasets, such as those in the Simons project. Our threshold for allele counts in the region where it is “absent” was < 10 (<0.1% frequency), while in the rest of the world, we used a cutoff of > 1000 counts (>25% allele frequency overall). We explored RACA only for Africa, Europe, and East Asia. The high degree of admixture among Native American populations made RACA calculation impossible. Because of strong admixture between South Asian and European populations, we computed RACA only for European populations and omitted South Asian populations. The data on RACA SNPs are shown in Table 2. It should be noted that biological interpretation of RACA is ambiguous. RACA may appear not only due to purifying selection of this allele, but also via fixation of the opposite allele through positive selection.

East Asia has the highest number of RACA SNPs compared to the other regions (Table 2). This observation is in a sharp contrast to the number of ARSA SNPs, where East Asia has the fewest of all regions. The observed enrichment of RACA SNPs in EAS may be explained by the recent bottlenecking of East Asian populations coupled with stronger genetic drift as compared to European populations [26]. In this case, many alleles could have been lost due to random genetic drift. This hypothesis is supported by the notion that East Asian populations have the least genetic diversity [27,28]. The last column of Table 2 shows the proportion of ancestral versus derived alleles among RACA. African RACA have the lowest proportion (3%) of ancestral alleles, while European RACA have the opposite trend (88% of ancestral alleles). The number of independent (no linkage disequilibrium) European RACA with known ancestral/derived allele status may be as small as 8, and thus their highest proportion of ancestral alleles (88%) is a very rough estimation without appropriate statistical support. Nonetheless, our Monte Carlo simulations on 8 outcomes demonstrated that European RACA ancestral alleles exceed RACA-derived alleles (*p*-value = 0.03). Hence, RACA ancestral/derived status is drastically different between Africa and other studied regions (Europe and East Asia).

### 3.3. Previously Annotated RACA and ARSA SNPs

The BioMart data mining tool [29] was used to find SNPs with previously associated phenotypes among our sets of ARSA and RACA SNPs. All ARSA SNPs from column 4 in Table 1 and RACA SNPs from column 2 in Table 2 were examined. BioMart sources data from the following databases: AMDGC, ClinVar, dbGaP, GEFOS, GIANT, HGMD-PUBLIC, MAGIC, NHGRI-EBI GWAS catalog, and Teslovich. These already annotated ARSA and RACA SNPs are presented in Appendix A and are summarized in Table 3. It shows the total number of SNPs and the subset for which substantial statistical power (*p*-values < 10^−5^) have been reported. Because a majority of GWAS and other studies of SNP association with disease and/or physiological conditions use chip microarrays, containing limited subsets of known SNPs, the intersection of our whole-genome ARSA and RACA SNP sets with BioMart output cannot be large. European populations were most frequently studied in GWAS; therefore, they are most densely represented in Table 3, while Oceania populations have the least representation.

### 3.4. Analysis of Possible Functions of RACA and ARSA SNPs

SNPs that undergo rapid increase in their allele frequencies may have certain functional significance in geographical regions where they underwent quick change. Keeping this in mind, possible roles of ARSA and RACA SNPs for biological significance were examined. The genomic context of ARSA and RACA SNPs were determined using snpEff5.1 software. The list of genes associated with ARSA and RACA SNPs for different geographical regions are presented in Appendix A. Figure 4A summarizes the number of genes associated with ARSA SNPs in African, American, East Asian, European, and Oceanian regions and their intersections. Figure 4B represents analogous gene distribution of African, East Asian, and European RACA SNPs. Distributions of ARSA and RACA SNPs per gene are shown in Figure 5A,B, respectively. There is strong statistically significant intersection above random expectation of genes associated with ARSA in all regions (Figure 4A). Note that in Africa, there are many more genes (6400) associated with ARSA than in any other region. Assuming the total number of human genes is 25,000, then in a random sampling of 100 genes, about 25 of them should match the African set of 6400 genes (100 × (6400/25,000)) by chance. Therefore, a quarter of our set of ARSA-associated genes from America, East Asia, Europe, or Oceania should intersect African genes by chance. However, Figure 4A demonstrates that 58% of American and 55% European ARSA-associated genes intersect the African set of genes. Moreover, 45% of Oceania and 37% of East Asian genes match the African set. Furthermore, 60 ARSA-associated genes are common among Africa, America, and Europe. Finally, 10 ARSA-associated genes are common for four populations (see Figure 4A). Monte Carlo simulations demonstrated that all these intersections are larger than random expectation with *p*-values less than 10^−4^. The list of 97 ARSA-associated genes that intersect for three and four regions are described in Appendix A. This set of 97 genes containing ARSA from three or four different regions may be the most interesting because it experienced multiple independent allele propagations. Among these 97 genes, 14 are non-coding RNAs or predicted proteins with unknown functions. Among the remaining known 83 genes, the dominant fraction of 35 genes (or 42%) are related to neural system and neurological disorders including two genes related to speech development (FOXP2 and CNTNAP2, see Appendix A). Finally, Figure 4B demonstrates that there is the same statistically significant intersection above random expectation of genes associated with RACA genes between Africa and East Asia.

To gain insight into biological processes and functions associated with the region-specific variation, the sets of ARSA-associated genes from Appendix A were subjected to functional annotation analysis using the DAVID web resource [30]. Since African ARSA SNPs are too numerous (77,820), we made a top 10% subset of them, keeping 7782 SNPs with the highest frequencies of ARSA alleles. Overall, using DAVID, we analyzed 1138 (top 10%) African, 1259 American, 119 East Asian, 514 European, and 145 Oceanian ARSA-associated genes. The summary of this analysis is illustrated in Table 4. It includes all the biological processes, molecular functions, and pathways significantly enriched for ARSA genes in at least one region. The maximum shared annotations were found between Africa and America. Another point is the abundance of annotations related to development and functioning of the nervous system that correlates with the above results of functional evaluation of the subset of ARSA genes shared by multiple populations.

It is also known that the specific patterns of allele frequencies could be the result of natural selection. Due to small sample sizes, we could not perform common tests to detect signals of natural selection among our ARSA and RACA SNPs. We therefore turned to alternative sources of insight, particularly to published sets of genes showing evidence of natural selection. Four such gene sets named by us as set 1 (1995 genes [31]), set 2 (273 genes [32], set 3 (1365 genes [33]), and set 4 (4172 genes [34]) were intersected with the lists of ARSA- and RACA-associated genes (Table 5). The results of intersection demonstrate that only 5–10% of the annotated genes were in common with the data of relatively large sets 1, 3, and 4. The smallest intersection was between our data and set 2, which contained data from only three populations of 1000 Genomes.

In summary, the set of ARSA and RACA SNPs includes variants that could have evolved due to selection. These SNPs could influence general biological processes, and, in particular, the neural system.

## 4. Discussion

Among the three billion nucleotides of the human genome, there is no single nucleotide position where every member of one continental group has a particular nucleotide while everyone else from other continents have another nucleotide(s). This observation supports a recent common origin of all modern humans from remote regions and can confound the characterization of genetic differences between populations. In this paper, 84,799 ARSA and 17,446 RACA SNPs were discovered that help characterize the major genomic differences between geographically disparate populations. The most dramatic genomic differences are observed in ARSA SNPs where genomes of people from one continental region show an abundance of an allele at a particular chromosomal position, while in other continental regions, this allele is absent. The distribution of ARSA SNPs in continental regions offer some interesting insights (Table 1). The highest number of ARSA SNPs was observed in Africa, which supports the “Out-of-Africa” theory of the origin of humankind. Surprisingly, the lowest number of ARSA SNPs was observed in East Asia, rather than in America, as might be expected. Moreover, the number of ARSA SNPs in America exceeded that of even Europe. Another prominent type of regional-specific alleles are RACA SNPs, which were examined for Africa, Europe, and East Asia (Table 2). Surprisingly, the highest number of RACA SNPs were observed in East Asia, the region that has the fewest number of ARSA SNPs. Moreover, the number of RACA SNPs in Europe was found to be extremely low, while European ARSA SNPs were much more abundant. We do not have solid explanation of this phenomenon yet.

We also observed uneven proportions of ancestral versus derived alleles of ARSAs and RACAs in different regions (Table 1 and Table 2, last columns). African ARSAs have the highest frequency of ancestral alleles, while African RACAs have the highest frequency of derived alleles. This phenomenon does not seem compatible with the mainstream conception of human evolution. It might be associated with ancient admixture of prehistoric human populations. An example of such possible events is the well-accepted admixture of Neanderthals with pre-historic people, which occurred unevenly in different continents with the peak of admixture in Oceania [35,36]. Several schemes of ancient admixture have been proposed [17]. Alternative explanations of our results may also exist. For example, the biological reason for the origin of RACA may not be in purifying selection, but in the fixation of the opposite allele, which may be beneficial. In this view, like for ARSA SNPs, the derived alleles opposite to RACA alleles are more frequent in non-Africans, while in Africa, their rate was found to be substantially lower. This correlates with the results of Bergström et al. (2020) [15] that demonstrated that the proportion of ancestral alleles increased among high-frequency private African alleles, including those which were fixed. This enrichment of ancestral alleles could be reflection of their age. All in all, the observed peculiarities of ARSA and RACA distributions testify that the origin of people may be more complicated than previously proposed [37].

Fewer ARSAs were found in Europe, while America had more ARSA SNPs. This correlated with the data by Bergström et al. (2020) [15] and could reflect higher isolation of America from the main roads of human migrations and admixture during last 10,000 years that in turn did not promote accumulating similar private SNPs to Eurasian regions [38]. One exception here was Oceania that, like the Americas, was also in relative isolation. This effect could be due to specificity of the filtration protocol that did not include appropriate Oceania populations at the second step (compare step 1 and 2 in Table 1).

Comparing distribution of ARSA-associated genes among populations demonstrated that the substantial portion was shared between populations. ARSA-associated genes shared by multiple populations seemed to be crucial for human evolution across the globe. Other ARSA-associated genes could be associated with local differences in adaptation and demography. The most intriguing are shared genes between Africa and America. America had the maximum numbers of both unique and shared genes. This could be the result of the complex history of the Americans as compared to other continents, particularly among South Americans, whose ancestors had to pass areas with very different climates, finally settling in territories located in the latitudes with climatic conditions such as those encountered by early humans in Africa. According to the canalization/de-canalization hypothesis, initial pathways formed in an organism by its relationships with the environment can be compromised when the organism moves out its adaptive niche [39]. It will require novel adaptations (involving new genes) to new conditions, particularly to survive in climates of high latitudes in northern Eurasia. Returning to low latitudes in America would require reconstruction (recanalization) of specific pathways, including the involvement of some of the genes that had been utilized in Africa. This idea is supported by the results of functional annotations (the biological process, molecular functions, and pathways) found in each group where the most similarity was observed between Africans and Americans (Table 4). In other groups, the identified functions seem to be less important (Europeans and East Asians) or not to be significantly modified compared to Africans (Oceanians).

Comparisons of lists of ARSA- and RACA-associated genes with the sets of genes known to be under natural selection showed no abundance of such genes, and their occurrence was 5 to 10% per geographic region. It correlates with the inferences of Choudhury et al. (2014) [9] who analyzed distribution of common private SNPs (MAF ≥ 5%) in African, East Asian, and European samples from the 1000 Genomes Project and observed no distinct evidence for selective sweeps connected with common population-specific SNPs. However, their results could have been due to low efficacy of iHS and PBS metrics chosen for testing of signals of natural selection in the datasets containing many SNPs of low-to-moderate allele frequencies [40,41]. Another reason why the proportion of such genes was small among ARSA- and RACA-associated genes could be smoothing (homogenization) of allele frequency due to the uniting of human samples from different locations in one group. Such uniting could reflect general tendencies but mask differences arising due to local adaptation [7]. Considering the results of enrichment analysis, such a tendency seems be realized in adaptive adjusting of processes of neuronal development and nervous system functioning (synaptic transmission and organization of the synapse) that has been suggested in previous studies [9,42,43]. Enrichment of ARSA-associated genes with the genes related to neurobiology (42%) suggests the impact of selection on brain function. Since genes related to the nervous system are generally longer than genes expressed in other tissues [44], these ARSA associations might be biased. The extra size of brain-specific genes results from the large introns inside them, which are due to the accumulation of repetitive elements and genomic duplications that do not have important biological functions [45]. We conjecture that it is unlikely that SNPs inside non-functional intronic sequences underwent rapid change of their frequencies and became a significant source of ARSA SNPs. On the other hand, the importance of neuronal genes in human differentiation and adaptation was demonstrated in analyses of the worldwide distribution of SNPs associated with psychiatric disorders (e.g., schizophrenia), which can be considered as forms of brain functioning that allowed humans to survive in changing environmental conditions [46,47].

A broad range of ARSA and RACA SNP–trait associations were identified for each continental region; however, some patterns were observed in certain continents (Appendix A). For instance, among the African ARSA SNPs, over 16% were associated with cardiovascular and weight-related risk traits including elevated lipoprotein a levels (rs9457986) [48], high HDL cholesterol, and high diastolic and systolic blood pressure [49]. These SNPs were observed among Africans who likely consumed a typical modern diet higher in processed carbohydrates and hydrogenated fats. It is possible that these SNPs associated with higher rates of cardiovascular disease were selected against as humans left Africa and adopted different diets, thus explaining their prevalence within the African continent. Moreover, among the African ARSA SNPs, there were three SNPs corresponding to mutations in SPTA1 associated with spherocytosis type 3 (rs16840450, rs35121052, and rs7547313), which leads to lower malaria parasitemia [50]. Only four ARSA SNPs were found in the database for East Asia, one of which was rs3805322, a variant in alcohol dehydrogenase 4 (ADH4) associated with an increased chance of esophageal cancer [51]. These SNP–trait associations seem to correspond well to common traits and diseases found within these continental regions. The most common ARSA SNPs found were relating to neurological traits, accounting for 7% of ARSA SNPs in Europe, 15% in Africa, 75% in East Asia, and 33% in Oceania. It should be noted only four and three ARSA SNPs were found to have database results for East Asia and Oceania, which is why the percentage of neurological SNPs is proportionally high. SNP–trait associations within the RACA SNPs did not have any perceivable patterns within continents; however, there were many SNPs associated with eye and skin pigmentation across all continents, which may reflect adaptations due to climate changes between continents.

## 5. Conclusions

Classification and analysis of population-specific ARSA and RACA SNPs provided new insights into worldwide genetic diversity, some aspects of which will require further studies on larger sized populations. Both ARSA and RACA SNPs could be valuable candidates for inclusion into novel studies of human population structure and evolution. Because these polymorphic sites are enriched with functional alleles having various adaptive roles, their inclusion should strengthen future GWAS projects, particularly those related to risk of neuropathology.

The strength of our bioinformatics project lies in the usage of three independent whole-genome sequence databases for computation and verification of ARSA SNPs. In particular, we are satisfied with the characterization of the list of 3348 America-specific ARSA SNPs despite high admixture in American populations. One limitation was the small size of the Oceania sample. Moreover, we were unable to calculate ARSA SNPs from the Middle East and Arctic regions because people from these areas are absent in the 1000 Genomes database and have considerable admixture with neighboring populations.

## Figures and Tables

**Figure 1 genes-13-01472-f001:**
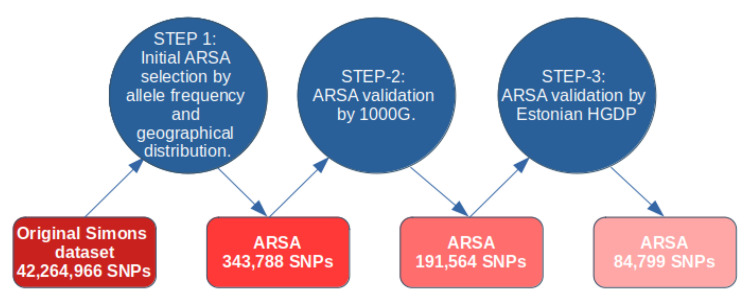
Legend: Computational methodology for characterization of ARSA SNPs with three validation points.

**Figure 2 genes-13-01472-f002:**
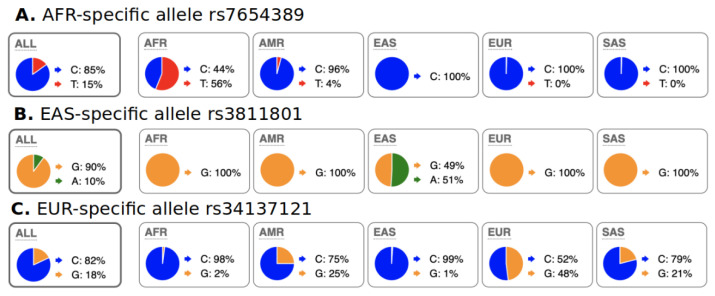
Examples of ARSA SNPs for Africa, East Asia, and Europe from population data of 1000 genome project as they presented on the Ensembl genome browser. A pie chart of regional SNP allele frequencies taken from the Ensemble website “Allele and genotype frequencies by population” (https://useast.ensembl.org/Homo_sapiens/Variation/ accessed on 20 May 2022) through SNP RS-identifiers. (**A**) Allele “T” is present only in Africa but absent in other continents (small frequency of “T” in Americas is due to admixture in 1000 Genomes populations). (**B**) Allele “A” is present only in East Asia and nowhere else. (**C**) Allele “G” is present in Europe but absent in Africa and East Asia. The presence of the “G” allele in Americas and South Asia (SAS) is due to admixture of Europeans with populations from these regions. African populations from the USA (ASW and ACB from 1000 Genomes) also have the “G” allele with 4 to 9% frequency that resulted in its 2% frequency in the total African sample. These ASW and ACB were excluded from our Step-2 calculations. Distribution of ARSA SNPs among continents is present in Table 1.

**Figure 3 genes-13-01472-f003:**
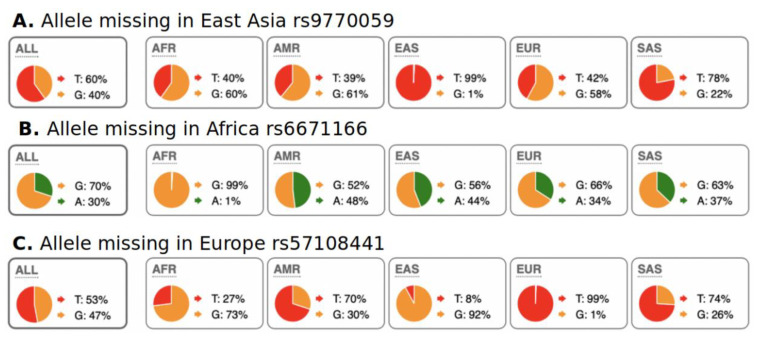
Examples of RACA SNPs for Africa, East Asia, and Europe. A pie chart of regional SNP allele frequencies taken from the Ensemble web site “Allele and genotype frequencies by population” (https://useast.ensembl.org/Homo_sapiens/Variation/ accessed on 20 May 2022) through SNP RS-identifiers. (**A**) Allele “G” is present globally, except for East Asia. (**B**) Allele “A” is present globally, except for Africa. (**C**) Allele “G” is present globally, except for Europe. One explanation for this phenomenon is fixation of common alleles in particular regions/continents. Distribution of RACA SNPs is shown in Table 2.

**Figure 4 genes-13-01472-f004:**
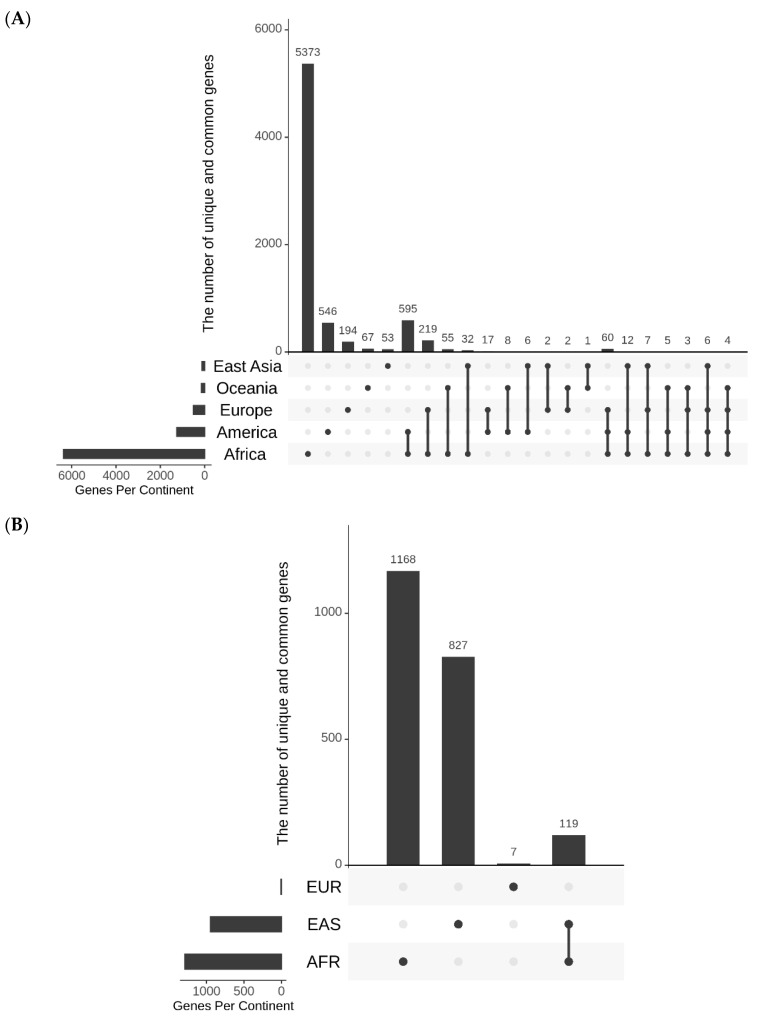
UpSet diagrams showing the number and nature intersections of gene lists for ARSA (**A**) and RACA (**B**). Vertical bars show the number of genes resulting from the intersections between gene lists. Horizontal bars depict the total number of genes in each continental group. The nature of intersections is shown with gray and black dots. A black dot means the presence of the gene in the corresponding continental group, while a gray dot means the absence of the gene in corresponding group. Black dots connected by lines indicate the continental groups involved in the interaction. For example, one black dot and four gray ones correspond to the vertical bar that shows the number of genes present in one continental group and absent in all other groups.

**Figure 5 genes-13-01472-f005:**
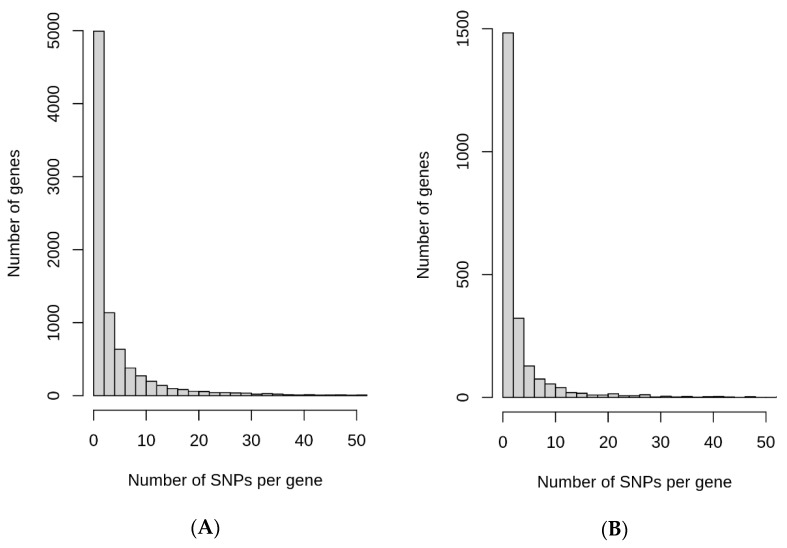
Distribution of the number of ARSA (**A**) and RACA (**B**) SNPs per gene.

**Table 1 genes-13-01472-t001:** Numbers of ARSA SNPs in five remote geographical regions.

Region	Step-1. Number of ARSA SNPs in the Simons Database	Step-2. Number of ARSA SNPs Filtered by 1000 Genomes	Step-3. Number of ARSA SNPs Filtered by EGDP Database	Step-4. Number of ARSA SNP Clusters	ARSA SNP Allele Status (Ancestral vs. Derived)
Africa	204,983	112,658	77,820	28,774	22% vs. 78%
Americas	46,994	4133	3348	3222	1% vs. 99%
East Asia	7789	441	362	272	7% vs. 93%
Europe	6585	2484	1911	1394	3% vs. 97%
Oceania	77,437	71,848 *	1358	453	4% vs. 96%

* Oceania populations are absent in 1000 Genomes; thus, the requirement for MAF > 18% is omitted for OCE at Step-2.

**Table 2 genes-13-01472-t002:** Numbers of RACA SNPs in three remote geographical regions.

Region	Number of RACA SNPs	Number of RACA SNPs Clusters	RACA SNP Allele Status (Ancestral vs. Derived)
Africa	6897	4159	3% vs. 97%
East Asia	10,524	3021	38% vs. 62%
Europe	25	16	88% vs. 12%

Note: Due to admixture of European, American, and Indian populations, the numbers may be biased, so Europe might be underrepresented. Nonetheless, European counts were much lower than African and EAS counts. The last column shows whether RACA is the ancestral or derived allele.

**Table 3 genes-13-01472-t003:** Number of SNPs from Table 1 and Table 2, with biological effects accessed with BioMart.

Region	# ARSA SNPs Total	# ARSA SNPs *p*-Value < 10^−5^	# RACA SNPs Total	# RACA SNPs *p*-Value < 10^−5^
Africa	353	191	382	362
America	17	5	N/A	N/A
East Asia	4	4	445	408
Europe	92	87	4	4
Oceania	3	0	N/A	N/A

**Table 4 genes-13-01472-t004:** Functional annotation of genes for ARSA SNPs obtained by the DAVID bioinformatics resource.

		AFR	AMR	EAS	EUR	OCE
**Biological processes**	Cell adhesion	8.8 × 10^−6^	0.003	+	+	
	Neurosciences	0.007	0.0004			
	Ion transport	0.007	0.005			
	Calcium transport	0.007	+	+	+	
	Transport	+	0.008			
	Potassium transport		0.021			
	Endosome	+	0.029			
	Golgi apparatus		0.02		0.05	
**Molecular functions**	Calcium channel	0.002	+	+		
	Ion channel	0.005	0.015			
	Guanine nucleotide releasing factor	0.009	0.015			
	Actin binding	0.014	+	+		
	Kinase	0.014	0.04			
	Serine/threonine-protein kinase	0.03	+			
	Voltage-gated channel	0.03	0.04		+	
	Potassium channel		0.015		+	
**Cellular component**	Cell junction	8.4 × 10^−10^	3.3 × 10^−7^		5.4 × 10^−4^	+
	Synapse	1.2 × 10^−6^	6.6 × 10^−5^	+	1.2 × 10^−2^	+
	Cell membrane	1.8 × 10^−4^	0.006		+	
	Cell protection	1.9 × 10^−4^	1.8 × 10^−4^	+	5.4 × 10^−4^	
	Cytoskeleton	0.002	+	+	+	
	Membrane	0.0024	0.032	+	0.035	
	Postsynaptic cell membrane	+	0.009			
	Cytoplasm	+	0.0025	+	0.049	
	Endosome	+	0.03			
	Golgi apparatus		0.02		0.049	
**Pathways**	Cortisol synthesis and secretion	0.007				
	cGMP-PKG signaling pathway	0.013	+		+	
	Tight junction	0.013				
	Axon guidance	0.025	0.0035			
	Arrhythmogenic right ventricular cardiomyopathy	0.026	0.025			
	Parathyroid hormone synthesis secretion and action	0.026	+			
	Type II diabetes mellitus	0.026				
	Calcium signaling pathway	0.026	+		+	
	Adrenergic signaling in cardiomyocytes	0.026	+		+	
	Circadian entrainment	0.028	+			
	Oxytocin signaling pathway	0.028	+			
	Cushing syndrome	2.80 × 10^−2^				
	MAPK signaling pathway	4.90 × 10^−2^	+			
	Long-term potentiation	4.90 × 10^−2^				
	Cholinergic synapse	+	6.30 × 10^−3^		+	
	Pathways in cancer	+	7.70 × 10^−3^			
	Glutamatergic synapse	+	1.20 × 10^−2^			
	Dopaminergic synapse	+	3.90 × 10^−2^			
	Insulin secretion	+	3.90 × 10^−2^			
	Inflammatory mediator regulation of TRP channels	+	3.90 × 10^−2^			
	Choline metabolism in cancer	+	3.90 × 10^−2^			
	Pancreatic secretion		5.00 × 10^−2^			

Note: Significant (*p* < 0.05) *p*-values after false discovery Benjamini correction for multiple testing are presented. Plus “+” denotes nonsignificant *p*-values, while the empty cells correspond to the biological process, molecular function, or pathway that were missing in the DAVID output for a specific region.

**Table 5 genes-13-01472-t005:** The number of ARSA- and RACA-associated genes shared with the published core sets of gens under natural selection.

Tab/Continental Group	SNPs	Set 1 (1995 Genes) [31]		Set 2 (273 Genes) [32]		Set 3 (1365 Genes) *		Set 4 (4172 Genes) **	
		n	*p*-Value	n	*p*-Value	n	*p*-Value	n	*p*-Value
Africa	ARSA	337	1	53	1	369	1.6 × 10^−2^	1171	2.2 × 10^−22^
America	ARSA	68	0.17	9	0.5	47	0.17	112	0.9
Europe	ARSA	12	0.93	5	0.21	10	0.93	39	0.21
East Asia	ARSA	15	5.6 × 10^−4^	4	0.02	4	0.87	12	0.87
Oceania	ARSA	10	0.1	0	1	2	1	9	1
AFR	RACA	115	1	16	1	143	5.3 × 10^−6^	349	1.6 × 10^−5^
EUR	RACA	2	0.71	0	1	3	0.31	5	0.31
EAS	RACA	89	0.83	10	0.83	80	2.8 × 10^−2^	242	3.4 × 10^−10^

* Genes from genome-wide published data (Appendix A from [33]). ** Genes from rawPophumanscanTable of the PopHumanScan catalog [34].

## Data Availability

All data and Perl programs are available on our website (http://bpg.utoledo.edu/~afedorov/lab/ATLAS3.html accessed on 20 May 2022) in a package that includes an Instruction Manual (ATLAS3instruction.docx) and Protocols (ATLAS3protocols.docx).

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
