# Peer review of "Analysis of Common SNPs across Continents Reveals Major Genomic Differences between Human Populations"

_genes, 2022, doi:10.3390/genes13081472_

Round 1
Reviewer 1 Report
Classification and analysis of population specific ARSA and RACA SNPs give us new insights into worldwide genetic diversity and this is important.
My suggestions are:
Abstract Line 11 "Common SNP represent ancient alleles ... ". I suggest:
" Common alleles tend to be more ancient than rare .."
Line 15: "17,446 cases" is not clear. Are these SNP's?
In methods - it is not clear how 18% MAF was calculated ?
Also, it is not clear how many samples they have analysed from every region; computation of ARSA and RACA is not described.
So, the chapter "2. Materials and Methods" has to be improved.
The methods are not adequately described.
Reviewer 2 Report
The study is in the thrust of very similar studies that have been published recently. I have the impression that the authors of this study did not test a specific hypothesis, but rather conducted a pure exploratory analysis.
I cannot fully agree with the authors' stated rationale for the chosen cutoff frequency for common SNPs. Their “biological” explanation for that is discussion about yin yang” haplotype pairs (what every this is) and that a higher cutoff frequency (20-25% MAF) often form haplotypes with very specific and intriguing properties (which would be what).
I was unable to review the section Material & Method as there are not much information given, difficult to read, and the given web-link http://bpg.utoledo.edu/~afedorov/lab/ATLAS3.html is not working. I also cannot check every perl script.
In Results, there are explorative enumerations of data. I cannot really comment on that. Figure 3 is incomprehensible to me.
The Discussion is very pure and often I miss a clear understanding in relating their finding to human evolutionary history/biology. Even reading the first sentence in the Discussion is already simply wrong. I have stopped reading the rest of the manuscript. I am sorry, but I cannot recommend the publication of this paper in its present form.
Reviewer 3 Report
I would like to thank the Authors of the Manuscript “What are the major genomic differences between human populations? Prominent SNP distinctions of people from Africa, America, East Asia, Europe, and Oceania” for the opportunity to provide commentary to their work.
To my understanding, this paper provides a frequency-based analysis of variants that are relatively common in a specific geographical region, but absent in all others (abundant region-specific alleles, ARSA), or variants that are common among all geographical regions, apart from one (regional absence of common alleles, RACA). This is performed with ad hoc programs produced by the Authors, with the rationale that locally common and locally depleted SNPs may be informative of human evolutionary signatures. Three whole-genome datasets have been used to detect and validate these variants and the resulting SNPs have been functionally supported through literature research and database mining. The geographically contextualized signatures reveal a preponderance of nervous system-related genes.
Overall, the paper is well-written, the content is very interesting, and the results are significant. The structure of the manuscript allows for a clear flow of information and each section is fleshed out beautifully.
I have relatively minor concerns:
1) The title may be a bit more exciting, as a list of geographical areas does not convey that much crucial information (e.g. it could be easily substituted with “worldwide”)
2) Please, could you justify the criteria for the choice of the specific populations of the Simons Genome Diversity Project and the rationale behind clustering them together (i.e. why do the East Asian populations count as a single group and not, for example, two)?
3) The same goes for the inclusion of ACB in the African population for the 1000 Genomes Project, but not ASW, given that both show signs of admixture with European ancestry (See The 1000 Genomes Project Consortium. A global reference for human genetic variation.Nature 526, 68–74 (2015). https://doi.org/10.1038/nature15393). The same reasoning goes for the South Asian populations as well.
4) Figures 1 and 2 are screenshots from the Ensembl website. I would appreciate if you could produce your own figure, so that the mouse pointer icon is not included.
5) Have you considered the possibility of a bias in the discovery of nervous system-related genes, given that they are overall the largest genes by size, and so it could be easier to find peculiar signatures in them?
